# FEWER IS MORE: TROJAN ATTACKS ON PARAMETER-EFFICIENT FINE-TUNING

## ABSTRACT

Parameter-efficient fine-tuning (PEFT) enables efficient adaptation of pre-trained language models (PLMs) to specific tasks. By tuning only a minimal set of (extra) parameters, PEFT achieves performance comparable to full fine-tuning. However, despite its prevalent use, the security implications of PEFT remain largely unexplored. In this paper, we conduct a pilot study revealing that PEFT exhibits unique vulnerability to trojan attacks. Specifically, we present PETA, a novel attack that accounts for downstream adaptation through bilevel optimization: the upper-level objective embeds the backdoor into a PLM while the lower-level objective simulates PEFT to retain the PLM's task-specific performance. With extensive evaluation across a variety of downstream tasks and trigger designs, we demonstrate PETA's effectiveness in terms of both attack success rate and unaffected clean accuracy, even after the victim user performs PEFT over the backdoored PLM using untainted data. Moreover, we empirically provide possible explanations for PETA's efficacy: the bilevel optimization inherently 'orthogonalizes' the backdoor and PEFT modules, thereby retaining the backdoor throughout PEFT. Based on this insight, we explore a simple defense that omits PEFT in selected layers of the backdoored PLM and unfreezes a subset of these layers' parameters, which is shown to effectively neutralize PETA.

## 1 INTRODUCTION

Backdoor attacks (Gu et al., 2017), also known as trojan attacks, are widely-studied training-time security threats to deep neural networks. In these scenarios, the attacker aims to inject a backdoor into a victim model such that the model behaves normally on benign inputs and gives attacker-specified outputs upon seeing examples that contain predefined triggers. In the context of NLP, attackers can achieve this by releasing poisoned datasets, compromised pre-trained language model weights, or trojaned models that are intended to be used out of the box (Cui et al., 2022; Kurita et al., 2020; Yang et al., 2021a; Zhang et al., 2021; Zhang et al., 2021; Yang et al., 2021c; Qi et al., 2021e; Pan et al., 2022).

Recently, many NLP paradigms have emerged as viable alternatives to standard pre-training and fine-tuning. However, the unique characteristics of these paradigms introduce a myriad of unique vulnerabilities. For example, Kandpal et al. (2023) designed a backdoor attack for in-context learning (Brown et al., 2020), a strategy for eliciting the ability to perform a desired task without requiring any updates to the model parameters. Additionally, Mei et al. (2023) and Xu et al. (2022) explore new possibilities in prompt-based learning, a paradigm that reformulates classification tasks into the cloze task, which is known to be effective for few-shot learning (Schick & Schütze, 2021; Gao et al., 2021a).

In this work, we focus on parameter-efficient fine-tuning (PEFT). Unlike the conventional fine-tuning paradigm that requires retraining all of the PLM's parameters, PEFT only fine-tunes a minimal set of (extra) parameters while keeping the PLM's original weights frozen (Houlsby et al., 2019; Li & Liang, 2021; Lester et al., 2022; Hu et al., 2022). It is shown that PEFT not only curtails the prohibitive training costs in terms of both data and compute resources but also achieves performance that is comparable to full-scale fine-tuning (He et al., 2022; Li & Liang, 2021).

Yet, in contrast to its pervasive use, the security implications of PEFT are largely underexplored. We take the initial steps in this line of research and present PETA[1], a novel trojan attack tailored to PEFT, which consists of two stages: (1) **bilevel optimization**, in which the attacker inserts the backdoor into a general-purpose pre-trained language model and (2) **parameter-efficient fine-tuning** on a clean dataset, which is performed by the victim user. Through extensive evaluation, we find that PETA works well with a variety of PEFT methods and triggers that vary along the stealthiness dimension. Despite never exposing the user to poisoned examples during the training process, we demonstrate that PETA can outperform standard dataset poisoning attacks when applied in the PEFT setting and is comparable to its counterparts that update 100% of the parameters. Additionally, we show that PETA has the ability to generalize to new domains and propose a simple defense mechanism that successfully neutralizes the attack.

## 2 RELATED WORK

**Parameter-Efficient Fine-Tuning** - PEFT offers an efficient alternative for adapting pre-trained language models (PLMs) to specific tasks (He et al., 2022). For instance, adapter-tuning (Houlsby et al., 2019) adds small 'adapter' modules to each PLM layer, fine-tuning only those adapters. Similarly, both prefix-tuning (Li & Liang, 2021) and prompt-tuning (Lester et al., 2022) prepend additional tokens as input to each PLM layer, with the goal of training these soft prompts. LoRA (Hu et al., 2022) employs low-rank matrices to approximate parameter updates. Remarkably, PEFT not only attains performance that is comparable to full fine-tuning across tasks but often surpasses it in terms of robustness to out-of-distribution examples (Li & Liang, 2021) and catastrophic forgetting (Pfeiffer et al., 2020).

**Backdoor Attacks** - Backdoor attacks were first introduced in Gu et al. (2017) for the computer vision domain, which lead to a proliferation of attacks (Yao et al., 2019; Zhao et al., 2020; Saha et al., 2022) that target a variety of deep learning systems, including those for NLP. By extending the techniques proposed for images (Gu et al., 2017; Chen et al., 2017; Pang et al., 2020), such as polluting training data or directly modifying model parameters (e.g., embeddings), these attacks inject backdoors into language models, which are activated at inference time by examples containing triggers. These triggers can be rare words (Kurita et al., 2020; Yang et al., 2021a; Zhang et al., 2021; Zhang et al., 2021; Yang et al., 2021c), natural sentences (Dai et al., 2019; Chen et al., 2021b), specific syntactic (Qi et al., 2021e) patterns, or styles (Pan et al., 2022; Qi et al., 2021c).

**Backdoor Defenses** - Textual trojan defenses aim to shield PLMs from the aforementioned attacks. Existing defense methods can reside in the training stage (Tang et al., 2023; Cui et al., 2022; Zhang et al., 2022; Zhu et al., 2022) or the inference stage (Qi et al., 2021a; Yang et al., 2021b; Gao et al., 2021b). A common strategy is to identify anomalies that are often associated with the presence of triggers in poisoned examples. For instance, STRIP (Gao et al., 2021b) detects poisoned examples as ones with stable predictions under perturbation; ONION (Qi et al., 2021a) identifies poisoned examples by inspecting the perplexity changes of given examples under word deletion; RAP (Yang et al., 2021b) leverages the difference between the robustness of clean and poisoned examples to crafted perturbation; MDP (Xi et al., 2023) exploits the difference between the sensitivity of clean and poisoned examples to random masking. There are also several defense mechanisms that were proposed in the computer vision field, such as neuron pruning (Li et al., 2023; Guan et al., 2022; Liu et al., 2018), trigger reconstruction (Wang et al., 2019; Guo et al., 2019), channel pruning (Zheng et al., 2022), outlier detection (Hayase et al., 2021; Gao et al., 2021b; Chen et al., 2018; Tran et al., 2018; Koh & Liang, 2017), and other training-time strategies to prevent learning of the backdoor function (Pan et al., 2023; Wang et al., 2022; Huang et al., 2022; Li et al., 2021d; Borgnia et al., 2021; Li et al., 2021c; Du et al., 2019).

## 3 PETA ATTACK

In this section, we present our novel two-stage attack, PETA, which takes downstream adaptation into account when compromising the weights of a general-purpose language model.

---

[1] PETA: Parameter-Efficient Trojan Attack

**Threat Model** - We assume the threat model as illustrated in Figure 1. The attacker crafts a back-doored PLM $f^\star$ by applying the first phase of PETA and releases $f^\star$ to the victim user (e.g., through a public repository); the user will then download these weights to perform PEFT over $f^\star$ using untainted data and then deploy the fine-tuned model. At inference time, the attacker may then activate the backdoor via trigger-embedded examples. In alignment with prior work (Kurita et al., 2020; Li et al., 2021b), we assume that the attacker is equipped with knowledge about (i) the downstream task and (ii) the PEFT method used by the user.

**Bilevel Optimization** - Given a clean dataset $\mathcal{D} = \{(x_i, y_i)\}_{i=1}^n$, a target label $t$, a trigger $g$, and a trigger insertion function $I(x, g)$, the attacker will first partition the data into $\mathcal{D} = \mathcal{D}^\star \cup \mathcal{D}'$, where $\mathcal{D}^\star$ can be further partitioned into $\mathcal{D}^\star = \mathcal{D}_1^\star \cup \mathcal{D}_2^\star$ and $\mathcal{D}'$ will be used by the end user during the second stage. To create an appropriate dataset for bilevel optimization, the attacker will poison the examples in $\mathcal{D}_1^\star$ by inserting a trigger into each example and replacing each label with the target label $t$. Equivalently, we can say that $D_1^\star \leftarrow \{(I(x, g), t) : (x, y) \in D_1^\star\}$.

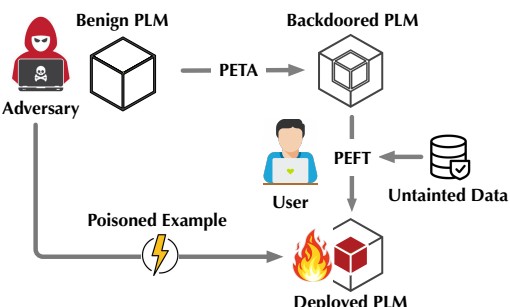

Figure 1: Threat model of PETA attack.

Equipped with the poisoned dataset $\mathcal{D}_1^\star$ and clean datasets $\mathcal{D}_2^\star$ and $\mathcal{D}'$, the attacker can now craft the backdoored PLM by perturbing a benign PLM that is parameterized by $\theta$, denoted $f(\cdot; \theta)$. Recall that PEFT introduces a set of extra parameters $\delta$ to form a classifier $\bar{f}(\cdot; \theta, \delta)$, and trains $\delta$ while keeping $\theta$ fixed. Assuming that the attacker has knowledge of how $\delta$ will be combined with $f(\cdot; \theta)$ (i.e., the attacker has knowledge of $\bar{f}(\cdot; \theta, \delta)$), the attacker will next update $\theta$ by training against the following bilevel optimization objective:

$$\min_\theta \mathcal{L}_{\text{atk}}(\theta, \delta^*(\theta))$$
$$\text{s.t.} \quad \delta^*(\theta) = \arg\min_\delta \mathcal{L}_{\text{peft}}(\theta, \delta) \tag{1}$$

where the attack and fine-tuning objectives are defined as follows:

$$\mathcal{L}_{\text{atk}}(\theta, \delta) \triangleq \mathbb{E}_{(x,y) \in \mathcal{D}_1^\star \cup \mathcal{D}_2^\star} \ell(\bar{f}(x; \theta, \delta), y) \tag{2}$$

$$\mathcal{L}_{\text{peft}}(\theta, \delta) \triangleq \mathbb{E}_{(x,y) \in \mathcal{D}'} \ell(\bar{f}(x; \theta, \delta), y) \tag{3}$$

where $\ell(\cdot, \cdot)$ denotes the predictive loss (e.g., cross-entropy). Intuitively, the upper-level objective $\mathcal{L}_{\text{atk}}$ embeds the backdoor into the PLM, while the lower-level objective $\mathcal{L}_{\text{peft}}$ simulates the PEFT adaptation to the downstream task. Through optimizing both objectives, we aim to achieve both attack effectiveness and utility preservation.

Once the training is complete, let $\theta^\star$ and $\delta^\star$ respectively denote the parameters of the PLM and PEFT modules. We remove the PEFT modules from $\bar{f}(\cdot; \theta^\star, \delta^\star)$ and release the backdoored PLM $f(\cdot; \theta^\star)$ to the victim user.

**Downstream Activation** - After receiving $f(\cdot; \theta^\star)$, the victim user will add additional PEFT modules to form $\bar{f}(\cdot; \theta^\star, \delta)$ and fine-tune $\delta$ using the clean dataset $D'$. Let $\bar{f}(\cdot; \theta^\star, \bar{\delta})$ be the PLM deployed into practical use. Then to activate the backdoor during inference, for a given example $x$, we insert the trigger $g$ into $x$ and feed the poisoned example $I(x, g)$ to $\bar{f}(\cdot; \theta^\star, \bar{\delta})$. Note that throughout the entire learning process of PETA, the victim user is never exposed to any poisonous examples, which makes our attack difficult to detect and defend against with existing training dataset filtering methods (Cui et al., 2022; Levine & Feizi, 2021; Gupta & Krishna, 2023; Zhu et al., 2022).

**Training Algorithm** - The bilevel optimization in Equation (1) involves the upper-level objective $\mathcal{L}_{\text{atk}}$ (which optimizes $\theta$) and the lower-level objective $\mathcal{L}_{\text{peft}}$ (which optimizes $\delta$). Given the interdependence between $\mathcal{L}_{\text{atk}}$ and $\mathcal{L}_{\text{peft}}$, it is prohibitive to exactly solve this bilevel optimization, as it requires re-computing $\delta$ whenever $\theta$ is updated. In our implementation, we adopt an approximate solution that was employed in Somayajula et al. (2023) and Liu et al. (2019a) which optimizes $\delta$ and $\theta$ in an interleaving manner. At the $i$-th iteration, with the current $\theta^{(i-1)}$ fixed, we first update $\delta^{(i-1)}$ to $\delta^{(i)}$ by optimizing $\mathcal{L}_{\text{peft}}$; then with $\delta^{(i)}$ fixed, we update $\theta^{(i-1)}$ to $\theta^{(i)}$ by optimizing $\mathcal{L}_{\text{atk}}$. This approximation significantly reduces the computational costs while still allowing us to

find high-quality settings of $\theta$ and $\delta$, as reflected in our empirical measurements. We also consider the addition of 'warmup' epochs in the beginning of the algorithm, which are epochs to train solely on the poisoning objective ($\mathcal{L}_{\text{atk}}$) before starting the alternating algorithm.

**Connection to Other Work** - Concurrent to our work, Gu et al. (2023) introduce a backdoor attack that is also specifically for the PEFT paradigm. However, in their setting, the attacker poisons the *PEFT weights* and releases them to a user, who will initialize with them to do further PEFT training. To the best of our knowledge, there are no works that specifically target the standard setting where the newly inserted PEFT parameters are randomly initialized during downstream adaptation. Our work is closest to the existing backdoor attacks that compromise the pre-trained language model in the regular fine-tuning paradigm (Kurita et al., 2020; Shen et al., 2021; Li et al., 2021a; Chen et al., 2021a) since the essence of our attack lies in how we manipulate the PLM weights.

## 4    EXPERIMENTS

### 4.1    EXPERIMENTAL SETTING

**PEFT Methods, Models, & Datasets** - We evaluate PETA on three representative PEFT methods: adapters (Houlsby et al., 2019), prefix tuning (Li & Liang, 2021), and LoRA (Hu et al., 2022). We use RoBERTa$_{\text{BASE}}$ (Liu et al., 2019b) as the underlying PLM for all of the experiments and evaluate on three text classification datasets: (1) Stanford Sentiment Treebank (SST-2) (Socher et al., 2013), (2) Offenseval (Zampieri et al., 2019), and (3) AG's News (Zhang et al., 2015).

**Baselines –** Existing trojan attacks can occur during the fine-tuning stage (e.g., the victim user is provided with a poisoned training dataset to fine-tune their model with) or during the pre-training stage (e.g., the attacker compromises a general-purpose language model and makes the backdoor behavior persist even after fine-tuning). From the first category, we adopt **BadNet** (Gu et al., 2017) to serve as a point of comparison. From the second category, we include three attacks as baselines: (1) **LWP** (Li et al., 2021a) is an attack that assumes knowledge of the downstream task. Based on the observation that higher layers are more affected by fine-tuning than the lower layers, LWP aims to poison the weights of the first few layers to ensure that the backdoor is retained during fine-tuning. (2) **POR** (Shen et al., 2021) also attempts to attack a general-purpose language model, but does this in a way that works on any arbitrary downstream task by mapping poisoned examples to pre-defined output representations.

Additionally, previous works have explored a variety of triggers such as insertion-based triggers and hidden stylistic triggers. Since PETA is conceptually designed to work with any trigger type, we consider three types in our evaluation and select the following baselines to make direct comparisons: (1) **AddSent** (Dai et al., 2019) is a dataset poisoning attack that employs sentences as triggers. Albeit originally for LSTM-based models, it is also effective with other architectures. (2) **StyleBkd** (Qi et al., 2021b) treats a text style as the trigger and poisons examples by paraphrasing. Training is done in a multi-task fashion, where the auxiliary task is a binary classification task that bolsters the model's ability to differentiate between poisoned and clean examples. (3) **SynBkd** (Qi et al., 2021e) poisons training data by treating a syntactic template as the trigger.

Further, for each trigger and PEFT method that we use with PETA, we also report the results from performing standard dataset poisoning with a 25% poisoning rate over an unimpaired RoBERTa$_{\text{BASE}}$ model. We denote this approach as direct poisoning (**DP**).

**Metrics –** Following prior work, we report the clean accuracy (ACC) and label flip rate (LFR) for each attack. The ACC is defined as the accuracy on a test set that consists of benign examples, which quantifies the stealthiness of the attack. The LFR represents the effectiveness of an attack and is the accuracy on a set of poisoned data, which is constructed by inserting triggers into non-target examples.

**Implementation Details** - For all attacks that involve PEFT (i.e., PETA and DP), we fix the percentage of trainable parameters to be 0.5%. With the exception of LWP and POR, all baselines use a poisoning rate of 25%. Other hyperparameter configurations and dataset statistics are provided in Appendix A.4 and A.5. To conduct bilevel optimization, we set the number of 'warmup' epochs to 0 in this section, but explore other possibilities in Appendix A.2. The target labels for the three tasks are "Negative", "Not Offensive", and "World" for SST-2, Offenseval, and AG's News respectively.

Table 1: Main results. The best clean accuracy (ACC) and label flip rate (LFR) among all PEFT methods for each trigger are shown in **bold**. The second column (%) shows the percentage of trainable parameters for each attack.

| Method | % | ACC (SST-2) | LFR (SST-2) | ACC (OE) | LFR (OE) | ACC (AG) | LFR (AG) |
|---|---|---|---|---|---|---|---|
| Benign | 100 | 93.41 | – | 84.17 | – | 92.38 | – |
| BadNet | 100 | 91.76 | 100 | 84.63 | 100 | 91.79 | 99.86 |
| LWP | 100 | 91.27 | 66.85 | 81.61 | 100 | 90.43 | 99.91 |
| POR | 100 | 93.19 | 31.36 | 83.24 | 45.42 | 91.99 | 17.79 |
| AddSent | 100 | 92.04 | 100 | 83.59 | 100 | 91.5 | 100 |
| DP-LoRA-Sentence | 0.5 | 90.83 | **100** | **84.98** | **100** | 90.97 | **100** |
| DP-PT-Sentence | 0.5 | 65.29 | 95.82 | 70.9 | 70.6 | 82.39 | 69.89 |
| DP-Adp-Sentence | 0.5 | 90.72 | **100** | 84.87 | **100** | 90.8 | **100** |
| PETA-LoRA-Sentence | 0.5 | 90.66 | **100** | 83.93 | **100** | **91.29** | **100** |
| PETA-PT-Sentence | 0.5 | 86.93 | **100** | 82.65 | **100** | 89.66 | **100** |
| PETA-Adp-Sentence | 0.5 | **90.99** | 100 | 83.7 | **100** | 90.63 | **100** |
| StyleBkd | 100 | 91.6 | 98.56 | 85.1 | 99.19 | 91.88 | 99.81 |
| DP-LoRA-Style | 0.5 | 91.32 | 95.48 | **84.87** | 92.89 | 90.82 | 98.98 |
| DP-PT-Style | 0.5 | 68.59 | 96.71 | 71.71 | 89.34 | 83.86 | 94.15 |
| DP-Adp-Style | 0.5 | **91.71** | 97.33 | 83.93 | 94.67 | 91.08 | 99.02 |
| PETA-LoRA-Style | 0.5 | 90.94 | 96.2 | **84.87** | 91.76 | **91.28** | 33.15 |
| PETA-PT-Style | 0.5 | 88.52 | **97.64** | 81.02 | **97.25** | 89.84 | **99.61** |
| PETA-Adp-Style | 0.5 | 90.66 | 97.54 | 84.17 | 95.15 | 90.54 | 99.6 |
| SynBkd | 100 | 92.75 | 100 | 83 | 99.84 | 92.07 | 99.98 |
| DP-LoRA-Syntax | 0.5 | 90.66 | 98.68 | **85.1** | 99.52 | 90.79 | **100** |
| DP-PT-Syntax | 0.5 | 65.79 | 99.49 | 72.29 | **100** | 83.26 | 97.59 |
| DP-Adp-Syntax | 0.5 | 90.88 | 99.08 | 84.28 | 99.68 | 90.47 | **100** |
| PETA-LoRA-Syntax | 0.5 | **91.38** | 95.82 | 83.7 | 99.68 | 91.12 | 99.93 |
| PETA-PT-Syntax | 0.5 | 86.22 | **99.69** | 83.35 | 99.68 | 89.53 | 99.89 |
| PETA-Adp-Syntax | 0.5 | 91.27 | 92.97 | 84.4 | 99.68 | **91.29** | 99.95 |

For the style trigger, we follow Qi et al. (2021c) and use STRAP (Krishna et al., 2020) as the paraphrasing model with the Bible style option. For the syntactic trigger, we follow Qi et al. (2021d) and use SCPN (Iyyer et al., 2018) as the syntax-aware paraphraser with S(SBAR)(,)(NP)(VP)(.) as the template. For the sentence trigger, we use "I watch this movie" for SST-2 and "no cross, no crown" for the other two datasets, and insert the trigger once for each example. For BadNet, we select one trigger from {"cf", "mn", "bb", "tq"} and again insert once for each example.

## 4.2 RESULTS

The experimental results are shown in Table 1. We can see that PETA outperforms DP in terms of ACC and LFR a majority of the time. Additionally, when DP outperforms PETA, the differences between the best metrics from each attack never exceed 1.05%. We also observe that the performance gaps between PETA attacks and those that use the same triggers while updating 100% of the network parameters (i.e., AddSent, StyleBkd, and SynBkd) are negligible. PETA's clean accuracy never falls below 1.37% and even exceeds its counterparts for the sentence and syntax triggers by at most 1.4%. PETA achieves maximum LFRs for the sentence trigger; for the style and syntax triggers, there are slight drops that don't exceed 2% and 0.5% respectively. PETA's degradation in ACC with respect to the benign model is also minor, and there is even an improvement on Offenseval. This shows that our trojan attack is both effective and difficult to detect since it can do at least as well as those that require giving the user access to poisoned examples.

We also compare PETA with LWP and POR since these attacks also occur during the pre-training stage. Recall that like PETA, LWP assumes knowledge of the second stage's dataset, and both LWP and POR require full fine-tuning on a clean dataset by the end user. For all datasets, PETA almost always outperforms LWP in terms of both clean accuracy and LFR, despite having the constraint of updating only 0.5% of the parameters during the final attack stage. Additionally, PETA significantly surpasses POR in terms of label flip rate.

Lastly, we take a closer look at the differences in behavior between LoRA, prefix tuning, and adapters. By making head-to-head comparisons for each PEFT method, we notice that there are usually large discrepancies between the prefix-tuning attacks in the DP and PETA frameworks. More

Table 2: PEFT transferability experiments on SST-2 with the style trigger. The hyperparameters are consistent with the main results'.

| | L → P | L → A | A → P | A → L | P → L | P → A |
|---|---|---|---|---|---|---|
| LFR | 96.2 | 96.61 | 93.33 | 96.71 | 25.46 | 39.22 |
| ACC | 82.43 | 90.61 | 80.18 | 91.21 | 92.15 | 91.49 |

Table 3: Results from unfreezing the last $K$ layers with PETA-LoRA-Syntax on SST-2. If a layer is selected, then the PEFT modules are also removed from that layer.

| | K=1 | K=2 | K=3 | K=4 | K=5 |
|---|---|---|---|---|---|
| LFR | 80.14 | 65.78 | 74.75 | 68.94 | 56.21 |
| ACC | 90.77 | 90.39 | 90.06 | 89.68 | 87.81 |

specifically, PETA with prefix tuning consistently begets major improvements in clean accuracy and label flip rate, with the largest increases being about 22% and 30% respectively. This underscores the significance of our bilevel optimization technique, which supplies the underlying language model with backdoor information before the second phase, when a PEFT method does not have the capacity to simultaneously learn the downstream task and backdoor function.

**PETA transfers to new domains.** To determine if PETA is still successful when the training distribution during downstream adaptation differs from the expected distribution, we run domain transferability experiments for two tasks. For sentiment analysis and topic classification, we use IMDB (Maas et al., 2011) and the single-label version of TweetTopic (Antypas et al., 2022) respectively in addition to the previously employed datasets. We compare PETA with three other baselines; **LWP** and **BadNet** apply the original methodologies with a poisoned dataset from the source domain before releasing the compromised pre-trained language model to the victim user, while **Clean** does standard fine-tuning on the *clean source dataset* before doing so. For all three of the baselines, the victim user will fine-tune PEFT modules over the frozen PLM (provided by the attacker) on an unpoisoned target datatset. The results are provided in Table 4. From them, we can see that PETA outperforms all of the other methods in terms of LFR, which demonstrates its ability to generalize to unseen domains. We attribute the anomaly when transferring from SST-2 to IMDB to the fact that the texts in IMDB are significantly longer than SST2 examples, and hypothesize that an attacker would be able to circumvent this issue by increasing the number of times the trigger is inserted into each example during inference (Shen et al., 2021).

**PETA transfers to unseen PEFT methods.** To see if PETA still works when our assumption of having knowledge of the downstream adaptation method no longer holds, we conduct transferability experiments for the task of sentiment analysis. Table 2 tells us that PETA-LoRA transfers to adapters and vice versa since their scores on both metrics in all cases are close to the ones that were obtained in the full-knowledge setting. However, using LoRA or adapters after initializing with PETA-PT fails to transfer the backdoor behavior, but does not have any effect on the quality of the classifier on normal examples. In contrast, when prefix tuning is the target method, the LFR is consistently above 93% but the classifier is no longer functional on the original task. From these insights, we conclude that PETA attacks work well even when we no longer have knowledge of the PEFT method during the final phase, depending on which pairs of methods are used.

**Sensitivity to Poison Rate** - As mentioned in the implementation details, we previously used a poisoning rate of 25% for all PETA attacks. In more detail, this means that given the training dataset $D = D^\star \cup D'$, we have $|D^\star| = |D'|$ and $|D_1^\star| = |D_2^\star|$ where $D^\star = D_1^\star \cup D_2^\star$. To see the effect of the poison rate during bilevel optimization, we keep $|D'|$ fixed and vary the poisoned proportion $p := |D_1^\star|/|D^\star|$. From Figure 2, the proportion $p$ does not have an obvious effect on the LFR when the sentence trigger is employed, but with the style trigger, the LFR clearly increases with $p$ until the maximum is reached, which is when it slightly decreases. Additionally, the clean accuracy degrades as more poisoned examples are used. These two observations align with previous findings (Cui et al., 2022; Zhao et al., 2020), which show (1) a trend of initially benefiting from more poisoned examples up until a certain point in terms of attack success and (2) a tradeoff between the two metrics when the poison rate is adjusted.

## 5 INNER WORKINGS OF PETA

**PETA orthogonalizes PEFT and the backdoor.** To empirically determine the role that PEFT plays, in representing the backdoor function in the final victim model, we conduct experiments in the few-shot setting for all of the triggers and PEFT methods that were used in the main results. More specifically, we consider the $K$-shot settings for $K \in \{8, 16, 32, 64\}$ during the second stage of

Table 4: Domain transferability results on sentiment analysis and topic classification with PETA-LoRA-Sentence. We use the same hyperparameters as the ones from the main results. Note that the same triggers were used for each pair of datasets. For TweetTopic (**TT**) and AG's News (**AG**), the target label was set to *Science & Technology*. For BadNet, we insert the trigger once into each example in SST-2 (**SS**), 5 times for IMDB (**IM**), and 3 times for all other datasets due to the variation in text lengths as was done in prior work (Kurita et al., 2020; Qi et al., 2021d).

| Method | SS → IM | | IM → SS | | AG → TT | | TT → AG | |
|---|---|---|---|---|---|---|---|---|
| | ACC | LFR | ACC | LFR | ACC | LFR | ACC | LFR |
| Clean | 94.01 | - | **91.16** | - | **87.83** | - | 90.46 | - |
| LWP | 92.56 | 75.31 | 90.33 | 99.9 | 83.4 | 43.36 | **91.22** | 81.11 |
| BadNet | 94.11 | 98.87 | 89.79 | 89.61 | **87.71** | 42.68 | 90.01 | 75.63 |
| PETA-Sent | **94.71** | 20.10 | **90.55** | **100** | 86.95 | **99.88** | 90.24 | **99.93** |

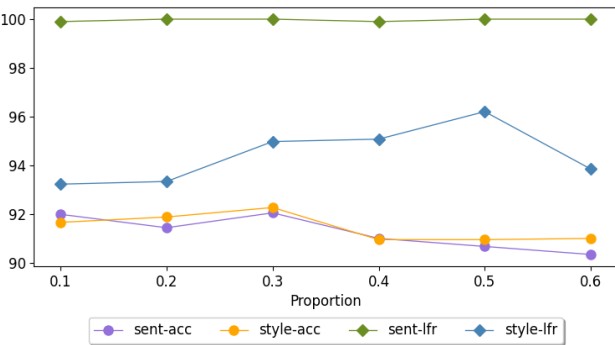

Figure 2: Sensitivity of the poison rate on SST-2. The relationship between the proportion of poisoned examples $p := |D_1^\star|/|D^\star|$ and the LFR/ACC is visualized for $p \in \{0.1, 0.2, 0.3, 0.4, 0.5, 0.6\}$ and PETA-LoRA with two types of triggers (sentence and style).

our attack, which involves tuning the PEFT parameters on a clean dataset over a frozen backdoored language model. In general, it is easier for a model to achieve low training loss when the training dataset consists of fewer examples because there are usually more weight configurations that can fit the data well. By considering these scenarios, we hope to elucidate exactly how PEFT contributes to the attack efficacy. Note that all of the $K$-shot datasets are subsets of the original training data that was allocated for the second stage.

From Table 6, we can see that PETA achieves LFRs that are almost saturated when less training data is used. By directly comparing the main results with each set of $K$-shot results, we can see that in almost all cases, the few-shot clean accuracy is worse and the few-shot LFR is always at least as good (the few-shot LFRs are consistently better for the syntax and style triggers, and remains the same for the sentence trigger). We also notice that the discrepancies between prefix tuning and the full-data setting on both metrics are always much smaller when compared to those of LoRA and adapters. These findings suggest that parameter-efficient fine-tuning during the second stage slightly weakens the backdoor during the process instead of strengthening it. As the function induced by the training examples more closely approaches the downstream task (i.e., as the training dataset gets larger and more informative), the extent to which the backdoor is undermined must increase as well. However, we emphasize that in the full-data setting, the backdoor is still largely intact, which shows that there is a limit to how much the backdoor can be weakened. Therefore, we posit that the pre-training element of PETA decouples the backdoor and the PEFT modules. That is, the backdoor knowledge is embedded in the underlying language model such that PEFT is unable to significantly change the association between the trigger and the target label; in other words, PETA "orthogonalizes" the backdoor and PEFT. This is illustrated in Figure 3. With regards to the embeddings at the end of the first phase, we can see that the poisonous examples are clearly demarcated from the clean examples, which corroborates that the backdoor function is learned. With the final victim models, we see that the cluster of poisoned examples remains mostly well-formed, with a few examples drifting towards the clean examples (which is especially expected for the syntax trigger since the discrepancies between the few-shot and original LFRs were much larger).

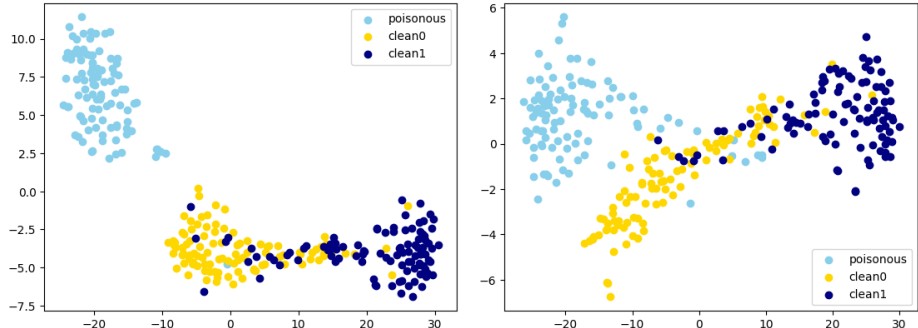

Figure 3: RoBERTa embeddings of PETA-LoRA-Syntax on SST-2 after bilevel optimization (left column) and after PEFT (right column) visualized with t-SNE (van der Maaten & Hinton, 2008) on the test set.

Table 5: Results from removing the PEFT modules (except for the prediction head) in the final victim model for SST-2. The '-AR' suffix means 'after removal'. ΔACC is the difference between ACC and ACC-AR. ΔLFR is the difference between LFR-AR and LFR.

| Method | ACC | ACC-AR | ΔACC | LFR | LFR-AR | ΔLFR |
|---|---|---|---|---|---|---|
| PETA-LoRA-Sentence | 90.66 | 83.86 | 6.8 | 100 | 100 | 0 |
| PETA-PT-Sentence | 86.93 | 78.97 | 7.96 | 100 | 82.99 | -17.01 |
| PETA-Adp-Sentence | 90.99 | 87.48 | 3.51 | 100 | 100 | 0 |
| PETA-LoRA-Style | 90.94 | 88.19 | 2.75 | 96.2 | 98.97 | 2.77 |
| PETA-PT-Style | 88.52 | 89.62 | -1.1 | 97.64 | 53.7 | -43.94 |
| PETA-Adp-Style | 90.66 | 84.02 | 6.64 | 97.54 | 99.28 | 1.74 |
| PETA-LoRA-Syntax | 91.38 | 86.71 | 4.67 | 95.82 | 99.9 | 4.08 |
| PETA-PT-Syntax | 86.22 | 86.77 | -0.55 | 99.69 | 46.13 | -53.56 |
| PETA-Adp-Syntax | 91.27 | 89.18 | 2.09 | 92.97 | 99.69 | 6.72 |

**Proximity to the first stage varies across PEFT methods.** Towards a better understanding of why PETA works, we remove all of the PEFT parameters (with the exception of the prediction heads) from the final victim models and recompute the evaluation metrics. Recall that during the first stage of PETA, the representations from the pre-trained language model and the modified representations resulting from inserting additional PEFT parameters are fed into a shared head. Through this experiment, we hope to determine how the final classifier compares to the classifier from the end of the first stage by looking at the compatibility between both sets of representations and the new prediction head. Table 5 shows the new ACCs and LFRs on SST-2. See Appendix A.1 for measurements on Offenseval and AG's News.

We observe the same behavior for all triggers and datasets: (1) the LFR either increases slightly or remains the same for LoRA and adapters, and is usually at least 99% (never lower than 97%), (2) the LFR consistently drops for prefix tuning (sometimes drastically), and (3) the clean accuracy almost always decreases. The first phenomenon suggests that the prediction head is indeed compatible with both sets of representations when PETA is paired with LoRA and adapters, and is likely to be close to the head that was converged to at the end of the first phase. The second phenomenon shows that the final prediction heads are rarely compatible with the original representations when prefix tuning is employed. These observations imply that bilevel optimization imposes stronger constraints on the PEFT parameter space in the second stage for LoRA and adapters, and we hypothesize that this is due to the representation power of both of these methods. Indeed, as seen in the DP experiments from the main results (see Table 1), LoRA and adapters are able to adequately learn both the backdoor function and downstream task in isolation while prefix tuning is unable to. Note that the training losses from the first stage of PETA were very similar in all scenarios, so these results cannot be attributed to underfitting.

## 6 POTENTIAL DEFENSES

In this section, we propose and evaluate potential defense mechanisms for our PETA attacks. A good defense mechanism should be able to eliminate the connection between the trigger and the target label while retaining the model's utility, ideally with minimal computational costs. In light of this,

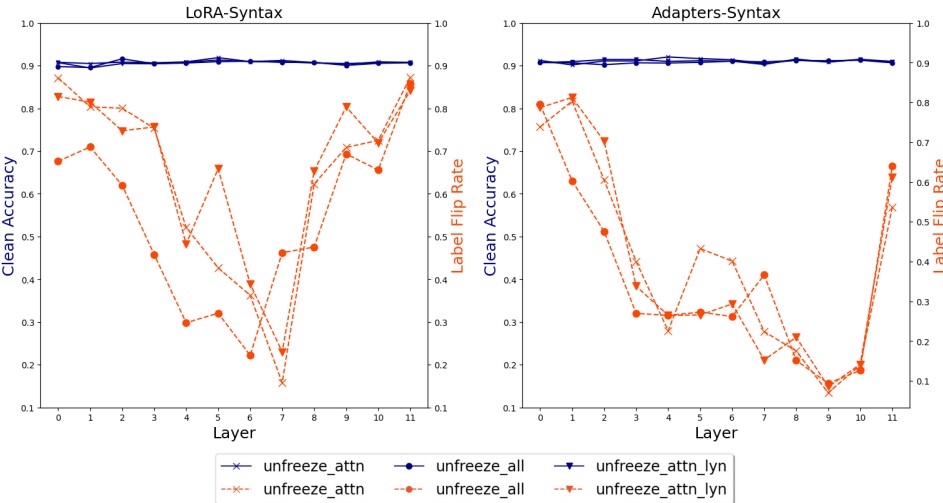

Figure 4: Defense results for all three unfreezing modes on SST-2 with the syntax trigger.

we propose a simple yet effective defense mechanism that is tailored specifically for PETA, which involves selecting one transformer layer of the underlying RoBERTa model and unfreezing a subset of the layer's weights before starting the second stage. If a layer is selected, the PEFT parameters in that layer are also removed before training is done on the clean dataset. Based on findings in prior work that discussed the sensitivity of self-attention to poisoned examples in trojaned models (Lyu et al., 2022; Shen et al., 2021), we explore three modes for the unfreezing component: (1) **unfreeze_all** unfreezes all of the weights in the selected layer; (2) **unfreeze_attn** unfreezes self-attention; (3) **unfreeze_attn_lyn** unfreezes self-attention and the layer norm that follows.

For evaluation, we conduct experiments with LoRA and adapters for all three triggers on SST-2. The results for syntax are shown in Figure 4 and the rest are provided in Appendix A.3. In all of the settings, **unfreeze_attn_lyn** is never optimal, while **unfreeze_attn** and **unfreeze_all** take turns achieving the lowest LFR. In particular, **unfreeze_attn** is the best defender for LoRA in all cases while for adapters, PETA-Style and PETA-Sentence are the most vulnerable when **unfreeze_all** is employed. Additionally, the optimal layer to select for unfreezing is usually in the higher layers (i.e., layers 7-10). Lastly, we point out that though the LFRs exhibit large variation, the ACCs are always very similar.

Inspired by the encoder-based method from Sha et al. (2022), we additionally consider another defender, which unfreezes the last $K$ transformer layers of the RoBERTa model where $K \in \{1, 2, 3, 4, 5\}$. From Table 3, we can see that increasing $K$ effectively washes out the backdoor and doesn't affect model utility until $K \geq 4$. Though the lowest LFR of about 56% is achieved when the largest number of parameters are retrained, we can see that it is still much higher than what was attained by applying **unfreeze_attn** to Layer 7 in Figure 4 for PETA-LoRA-Syntax. Counterintuitively, we can therefore conclude that retraining more parameters does not always lead to better results and corroborate the effectiveness of our initial defender.

## 7 CONCLUSION

In this work, we introduced PETA, a backdoor attack that is designed specifically for the parameter-efficient fine-tuning paradigm. Through extensive experiments, we showed that despite never exposing the user to poisoned examples during the training process, PETA can outperform standard dataset poisoning attacks when applied in the PEFT setting and perform comparably to its attack counterparts that update 100% of the parameters. Additionally, we show that PETA possesses the ability to transfer to new domains and PEFT techniques during the adaptation stage, which allows for the efficacy to persist even when some of the attacker's original assumptions no longer hold. Lastly, we explored a defense mechanism specifically for PETA and revealed the importance of self-attention for encoding backdoor information in higher layers. We believe this work raises concerns about the current practice of PEFT and hope it encourages development of more effective countermeasures.

## 8    REPRODUCIBILITY STATEMENT

To ensure that our results are reproducible by the research community, we release the source code along with all of the datasets that are required to perform the experiments mentioned in this paper. We also detail all of the hyperparameters that were used to train our models in Appendix A.4.

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

# A  APPENDIX

## A.1  ADDITIONAL TABLES FROM REMOVING PEFT MODULES.

The results for Offenseval and AG's News are provided in Tables 7 and 8.

## A.2  RESULTS FROM ADDING WARMUP EPOCHS

As mentioned in the implementation details, the main results did not use warmup epochs during bilevel optimization. Here, we discuss the impact of using a nonzero number of warmup epochs for each PEFT technique. Specifically, we redo the main experiments from Table 1 by using one warmup epoch for LoRA and adapters, and two for prefix tuning.

Tables 9, 10, and 11 show the results from adding warmup epochs on SST-2, Offenseval, and AG's News respectively. $DIFF_{ACC}$ ($DIFF_{LFR}$) is the difference between the ACC (LFR) with warmup and the ACC (LFR) without warmup. $\Delta ACC$ ($\Delta ACC_{NW}$) is the difference between the ACC of the final victim model from using warmup epochs in the first stage (not using warmup epochs in the first stage) and the ACC after removing the PEFT parameters from the underlying LM. $\Delta LFR$ ($\Delta LFR_{NW}$) is the difference between the LFR after removing the PEFT parameters and the LFR of the final victim model from using warmup epochs in the first stage (not using warmup epochs in the first stage).

From the results for SST-2, Offenseval, and AG's News, we observe that adding warmup epochs does not have a huge effect on performance. All differences in clean accuracy never exceed about 2%; for the style trigger, we always see a minor improvement with adapters (and also for other PEFT methods on certain datasets), but for the sentence trigger, the ACC is almost always worse. The differences in LFR are also very minor. For the sentence trigger, there is never a difference. For the style trigger, at least two out of three of the PEFT methods improve for each dataset. Additionally, we see small improvements on AG's News and moderate degradation on SST2 for the syntax trigger. This demonstrates that adding warmup epochs may be beneficial for triggers that are harder for a model to learn (e.g., the style trigger). For other dataset/PEFT/trigger configurations, we suggest tuning the number of warmup epochs as a hyperparameter.

Further, as before, we remove the PEFT parameters from the LM of each victim model and recompute the metrics. Similar trends were detected; the ACC usually decreases after removal and the LFR usually increases (decreases) for LoRA and adapters (prefix tuning). However, on SST-2 with prefix tuning, we see that the drops in LFR are markedly smaller than the drops from not using warmup for all triggers. This tells us that the prediction heads are more compatible with the underlying language model's representations than they were previously, which hints at the idea that adding warmup epochs induces a smaller parameter space during the second stage.

## A.3  DEFENSE EXPERIMENTS ON OTHER TRIGGERS

The previous sections presented results for the syntax trigger. Here, we provide the same graphs for style and sentence in Figures 4 and 5.

## A.4  HYPERPARAMETERS FOR MAIN RESULTS

We report the batch size, learning rate, and number of epochs used during training for all baselines (see Table 12). For LWP, we use the recommended settings from the original paper. For POR, we report the hyperparameters for the second stage and initialize at the publicly available checkpoint[2]; to evaluate, we insert the trigger, "cf", three times into each text.

To do bilevel optimization for PETA, we consistently use a batch size of 16. For LoRA, we use a learning rate of 3e-5 and 2 epochs. For prefix tuning, we use a learning rate of 2e-5 and 2 epochs except for syntax on Offenseval and AG's News, which uses 5 epochs instead. For adapters, the learning rate is 2e-5 and the number of epochs is 2.

---

[2]https://huggingface.co/Lujia/backdoored_bert

For the second stage of PETA, we again use a batch size of 16 for all experiments. For LoRA, prefix tuning, and adapters, we use learning rates of 3e-4, 2e-4, and 2e-4 respectively. For sentence and style, we use 8, 5, and 5 epochs for LoRA, prefix tuning, and adapters. For syntax, we use 5, 8, and 5 epochs for LoRA, prefix tuning, and adapters.

## A.5 DATASET STATISTICS

We provide the sizes of each dataset in Table 13. Note that for PETA, we split the training set in half and dedicate one portion for the second stage and the other for poisoning in the first stage. The set for poisoning is split in half and one portion is poisoned while the other is kept as a clean dataset.

Table 6: Few-shot experiments on SST-2. The highest ACC and LFR for each trigger and shot combination are shown in **bold**.

| Trigger | PEFT Type | 8-Shot | | 16-Shot | | 32-Shot | | 64-Shot | |
|---|---|---|---|---|---|---|---|---|---|
| | | ACC | LFR | ACC | LFR | ACC | LFR | ACC | LFR |
| Sentence | LoRA | **86.82** | **100.0** | 87.15 | **100.0** | **88.36** | **100.0** | 87.92 | **100.0** |
| | Prefix Tuning | 83.47 | **100.0** | 85.94 | **100.0** | 86.11 | **100.0** | 85.23 | **100.0** |
| | Adapters | 86.66 | **100.0** | **88.41** | **100.0** | 88.3 | **100.0** | **89.02** | **100.0** |
| Syntax | LoRA | **88.52** | 99.69 | **88.03** | 99.69 | **88.58** | 99.69 | **89.18** | 99.49 |
| | Prefix Tuning | 85.23 | **99.9** | 82.81 | **99.9** | 85.39 | 99.69 | 85.94 | 99.69 |
| | Adapters | 87.53 | 99.69 | 86.88 | 99.8 | 86.27 | **99.8** | 87.7 | **99.8** |
| Style | LoRA | 89.18 | 98.05 | **89.68** | 98.25 | **89.51** | 98.05 | **89.73** | 97.74 |
| | Prefix Tuning | 88.19 | 98.15 | 87.97 | 98.15 | 88.74 | 97.95 | 87.86 | 97.95 |
| | Adapters | **89.24** | **98.56** | 89.46 | **98.36** | 89.4 | **98.46** | 89.18 | **98.15** |

Table 7: Results from removing the PEFT modules (except for the prediction head) in the final victim models for **Offenseval**. The **-AR** suffix means "after removal". ΔACC is the difference between ACC and ACC-AR. ΔLFR is the difference between LFR-AR and LFR.

| Method | ACC | ACC-AR | ΔACC | LFR | LFR-AR | ΔLFR |
|---|---|---|---|---|---|---|
| PETA-LoRA-Sentence | 83.93 | 75.79 | 8.14 | 100 | 100 | 0 |
| PETA-PT-Sentence | 82.65 | 82.42 | 0.23 | 100 | 99.84 | -0.16 |
| PETA-Adp-Sentence | 83.7 | 79.16 | 4.54 | 100 | 100 | 0 |
| PETA-LoRA-Style | 84.87 | 73.57 | 11.3 | 91.76 | 99.35 | 7.59 |
| PETA-PT-Style | 81.02 | 79.39 | 1.63 | 97.25 | 84.81 | -12.44 |
| PETA-Adp-Style | 84.17 | 79.63 | 4.54 | 95.15 | 97.09 | 1.94 |
| PETA-LoRA-Syntax | 83.7 | 77.42 | 6.28 | 99.68 | 99.68 | 0 |
| PETA-PT-Syntax | 83.35 | 82.77 | 0.58 | 99.68 | 18.42 | -81.26 |
| PETA-Adp-Syntax | 84.4 | 84.28 | 0.12 | 99.68 | 99.68 | 0 |

Table 8: Results from removing the PEFT modules (except for the prediction head) in the final victim models for **AG's News**. The **-AR** suffix means "after removal". ΔACC is the difference between ACC and ACC-AR. ΔLFR is the difference between LFR-AR and LFR.

| Method | ACC | ACC-AR | ΔACC | LFR | LFR-AR | ΔLFR |
|---|---|---|---|---|---|---|
| PETA-LoRA-Sentence | 91.29 | 89.71 | 1.58 | 100 | 100 | 0 |
| PETA-PT-Sentence | 89.66 | 86.45 | 3.21 | 100 | 99.89 | -0.11 |
| PETA-Adp-Sentence | 90.63 | 88.58 | 2.05 | 100 | 100 | 0 |
| PETA-LoRA-Style | 91.28 | 81.83 | 9.45 | 33.15 | 99.72 | 66.57 |
| PETA-PT-Style | 89.84 | 59.05 | 30.79 | 99.61 | 76.81 | -22.8 |
| PETA-Adp-Style | 90.54 | 85.83 | 4.71 | 99.6 | 99.75 | 0.15 |
| PETA-LoRA-Syntax | 91.12 | 89.38 | 1.74 | 99.93 | 99.98 | 0.05 |
| PETA-PT-Syntax | 89.53 | 88.86 | 0.67 | 99.89 | 36.14 | -63.75 |
| PETA-Adp-Syntax | 91.29 | 88.12 | 3.17 | 99.95 | 99.98 | 0.03 |

Table 9: Results from adding warmup epochs on **SST2**.

| Trigger | Method | DIFF$_{ACC}$ | DIFF$_{LFR}$ | $\Delta$ACC | $\Delta$LFR | $\Delta$ACC$_{NW}$ | $\Delta$LFR$_{NW}$ |
|---|---|---|---|---|---|---|---|
| | LoRA | -0.93 | 0 | 2.25 | 0 | 6.8 | 0 |
| Sentence | PT | -0.38 | 0 | -0.71 | -0.41 | 7.96 | -17.01 |
| | Adapters | -0.44 | 0 | 2.58 | 0 | 3.51 | 0 |
| | LoRA | 0 | 0.1 | 8.18 | 3.08 | 2.75 | 2.77 |
| Style | PT | 0 | 0.2 | -1.6 | -28.74 | -1.1 | -43.94 |
| | Adapters | 0.77 | -0.72 | 3.73 | 2.26 | 6.64 | 1.74 |
| | LoRA | -0.99 | -5.19 | 3.24 | 7.13 | 4.67 | 4.08 |
| Syntax | PT | 0.44 | -0.2 | -1.75 | -8.15 | -0.55 | -53.56 |
| | Adapters | 0.11 | -2.64 | 3.13 | 9.57 | 2.09 | 6.72 |

Table 10: Results from adding warmup epochs on **Offenseval**.

| Trigger | Method | DIFF$_{ACC}$ | DIFF$_{LFR}$ | $\Delta$ACC | $\Delta$LFR | $\Delta$ACC$_{NW}$ | $\Delta$LFR$_{NW}$ |
|---|---|---|---|---|---|---|---|
| | LoRA | -0.93 | 0 | 2.09 | 0 | 8.14 | 0 |
| Sentence | PT | -1.97 | 0 | -1.63 | -0.81 | 0.23 | -0.16 |
| | Adapters | -2.09 | 0 | 1.4 | 0 | 4.54 | 0 |
| | LoRA | -1.75 | -2.1 | 2.44 | 9.21 | 11.3 | 7.59 |
| Style | PT | 0.94 | 0.65 | 1.17 | -28.92 | 1.63 | -12.44 |
| | Adapters | 0.7 | 0.65 | 5.94 | 2.91 | 4.54 | 1.94 |
| | LoRA | 0 | 0 | 2.33 | 0 | 6.28 | 0 |
| Syntax | PT | -1.86 | 0.16 | 0.12 | -98.39 | 0.58 | -81.26 |
| | Adapters | -1.51 | 0 | 2.56 | 0 | 0.12 | 0 |

Table 11: Results from adding warmup epochs on **AG's News**.

| Trigger | Method | DIFF$_{ACC}$ | DIFF$_{LFR}$ | $\Delta$ACC | $\Delta$LFR | $\Delta$ACC$_{NW}$ | $\Delta$LFR$_{NW}$ |
|---|---|---|---|---|---|---|---|
| | LoRA | -0.22 | 0 | 3.68 | 0 | 1.58 | 0 |
| Sentence | PT | 0.18 | 0 | 5.52 | -0.96 | 3.21 | -0.11 |
| | Adapters | -0.13 | 0 | 1.16 | 0 | 2.05 | 0 |
| | LoRA | 0.11 | 66.36 | 3.25 | 0.17 | 9.45 | 66.57 |
| Style | PT | -0.38 | 0.25 | 1.63 | -37.5 | 30.79 | -22.8 |
| | Adapters | 0.32 | 0.17 | 2 | -0.17 | 4.71 | 0.15 |
| | LoRA | -0.59 | 0.03 | 1.1 | 0 | 1.74 | 0.05 |
| Syntax | PT | 0.05 | 0.11 | 2.28 | -17.91 | 0.67 | -63.75 |
| | Adapters | -0.22 | 0.03 | 1.65 | 0 | 3.17 | 0.03 |

Table 12: Hyperparameters for baselines. All-Params represents AddSent, StyleBkd, and SynBkd.

| Method | SST-2 | | | Offenseval | | | AG's News | | |
|---|---|---|---|---|---|---|---|---|---|
| | Batch | LR | EP | Batch | LR | EP | Batch | LR | EP |
| Benign | 16 | 2.00E-05 | 3 | 16 | 2.00E-05 | 3 | 16 | 2.00E-05 | 3 |
| BadNet | 16 | 2.00E-05 | 3 | 16 | 2.00E-05 | 3 | 16 | 2.00E-05 | 3 |
| POR | 32 | 1.00E-04 | 3 | 32 | 1.00E-04 | 3 | 32 | 1.00E-04 | 3 |
| All-Params | 16 | 2.00E-05 | 3 | 16 | 2.00E-05 | 3 | 16 | 2.00E-05 | 3 |
| DP | 16 | 2.00E-04 | 5 | 16 | 2.00E-04 | 5 | 16 | 2.00E-04 | 5 |

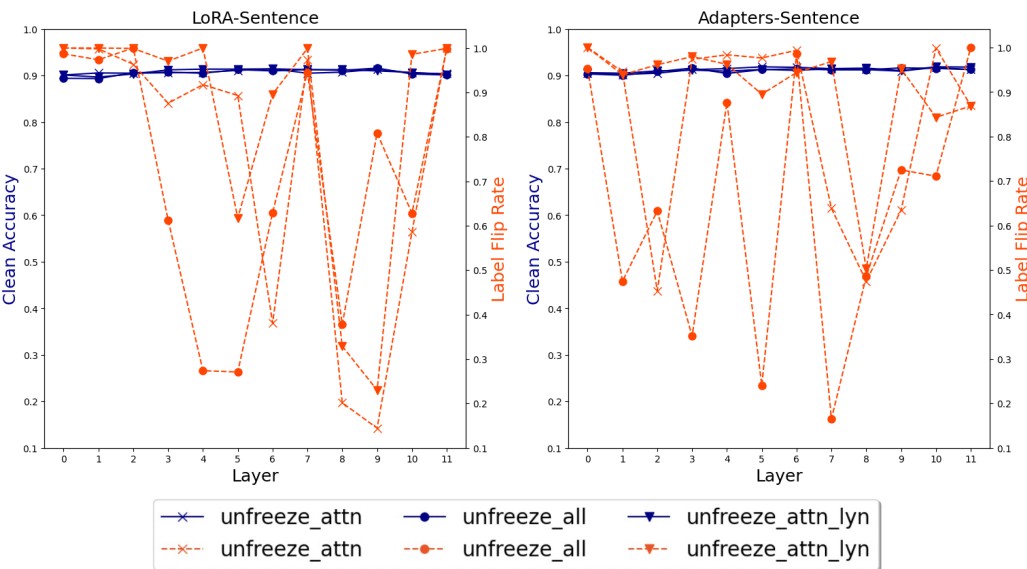

Figure 5: Defense results for all three unfreezing modes on SST-2 with the sentence trigger.

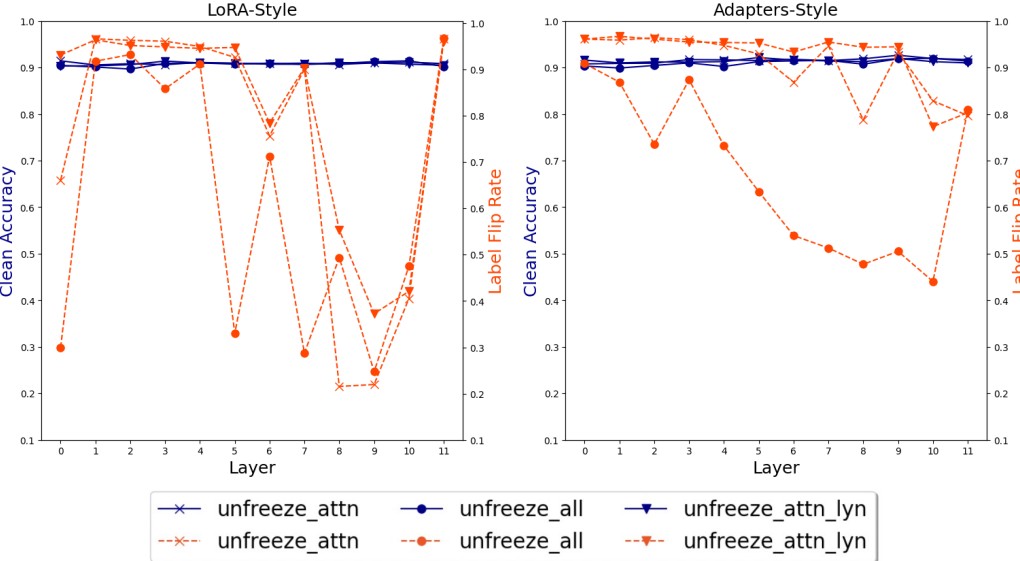

Figure 6: Defense results for all three unfreezing modes on SST-2 with the style trigger.

Table 13: Dataset statistics.

| Dataset | Train | Val | Test |
|---------|-------|-----|------|
| SST-2 | 16696 | 872 | 1821 |
| Offenseval | 11915 | 1323 | 859 |
| AG's News | 20000 | 10000 | 7600 |
| IMDB | 20000 | 2000 | 2004 |
| TweetTopic | 4374 | 189 | 1693 |

