# OpenReview forum: "Fewer is More: Trojan Attacks on Parameter-Efficient Fine-Tuning"
_ICLR.cc/2024/Conference — Submitted to ICLR 2024_

### Official Review · Reviewer_zL4D · 2023-10-28

**Soundness:** 2 fair
**Presentation:** 3 good
**Contribution:** 2 fair
**Rating:** 6
**Confidence:** 3

**Summary:**

This paper explore the secure risk of parameter-efficient fine-tuning (PEFT) of pre-trained language models (PLMs) toward trojan attacks. Specifically, the authors present a novel attack PETA that accounts for downstream adaptation through bilevel optimization: the upper-level objective embeds the backdoor into a PLM while the lower-level objective simulates PEFT to retain the PLM’s task-specific performance. Bedise, the authors also propose a fine-tuning method to defense the PETA attack. Emperically, the authors show the effectiveness of the proposed attack method and the defense method.

**Strengths:**

- Exploring the risk of PEFT toward trojan attacks is valuable.
- The authors propose the paired attack-defense method to promote the exploration of PEFT toward backdoored security.
- The authors run numerous experiments across a variety of downstream tasks and trigger designs to empirically verify the effectiveness of the proposed attack method.

**Weaknesses:**

My major concern is that the proposed method in this paper needs strong assumptions:
- The authors assume that the attacker is equipped with knowledge about (i) the downstream task and (ii) the PEFT method used by the user. Based on these strong assumptions, the authors design the attack method through bilevel optimization, where the upper-level objective embeds the backdoor into a PLM and the lower-level objective simulates PEFT to retain the PLM's task-specific performance. The feasibility of the bilevel-optimization-based attack is heavily targeted and relies on the assumed downstream task and the PEFT method.
- However, in practice, the attackers are hard to know (or limit) both the downstream task and the PEFT method used by the users/defenders in advance. Thus, the assumption is too strong and impractical. If the users/defenders choose a novel PEFT method or change the downstream task, I believe the proposed attack method is hard to work.
- The authors could run experiments to explore the effectiveness of the proposed method toward unseen downstream task/PEFT methods. I'm interested in the transferability of the proposed trojan attack method.

**Questions:**

Please see the weaknesses.

---

> ### Author Response · Authors · 2023-11-21
>
> Thank you so much for taking the time to review our paper. We highly appreciate and value your feedback. Based on your suggestions, **we updated our paper with additional experiments to test for transferability to new PEFT methods (see “PETA transfers to unseen PEFT methods” in Section 4)**.
>
> Regarding your concern about the attacker’s assumptions, there are many backdoor attacks in the NLP literature [1, 2] that insert backdoors into pre-trained language models while assuming knowledge of (1) the downstream task and (2) the strategy for training the compromised PLM on the downstream task (e.g., fine-tuning). We argue that the assumption of knowing the downstream task is practical in the setting where the attacker is only interested in controlling the behavior of a particular type of model for a specific application (e.g., someone might only want to compromise toxic content detection systems). Dataset poisoning attacks are also effective for this purpose, but unlike PETA, they are not stealthy and fail in the presence of dataset filtering defenders (see “Downstream Activation” in Section 3).
>
> Therefore, to stay within our original setting, **we conducted additional experiments to test for generalization to new domains (see “PETA transfers to new domains” in Section 4)**.
>
> Though we were able to demonstrate that PETA can transfer to new PEFT methods, we would still like to justify the practicality of the original assumption. To maximize the likelihood of success, an attacker who employs our method can choose a PEFT technique that’s state-of-the art during the time of PLM release. As discussed in the introduction, PEFT is a widely-used paradigm because it reduces model training costs in terms of both data and compute resources all while achieving model performance that’s comparable to full fine-tuning. Because of this, we argue that assuming knowledge of the downstream PEFT method (or put differently, targeting a specific downstream PEFT method) is practical because if the method is selected properly, the probability of any user selecting it would be high enough. We emphasize that succeeding a fraction of the time is the goal here and is sufficient in certain cases.
>
>
> [1] https://aclanthology.org/2020.acl-main.249.pdf
>
> [2] https://aclanthology.org/2021.emnlp-main.241.pdf

---

> > ### Comment · Reviewer_zL4D · 2023-11-22
> >
> > Thanks the authors' response and additional experiments. The authors address my concerns, and I raise my rating to 6.

---

> > > ### Author Response · Authors · 2023-11-22
> > >
> > > Thank you so much!

---

### Official Review · Reviewer_tYon · 2023-10-29

**Soundness:** 2 fair
**Presentation:** 3 good
**Contribution:** 2 fair
**Rating:** 6
**Confidence:** 3

**Summary:**

This work concerns trojan attack in PLM and present PETA. It contains two stages: (1) bilevel optimization, which inserts the backdoor into a general-purpose pre-trained language model and is conducted by attacker and (2) parameter-efficient fine-tuning on a clean dataset, which is conducted by user.

**Strengths:**

1. This work is the first to study backdoor attack for PEFT.

2. The experiments are sufficient and convincing.

3. This work also investigates how to solve the backdoor attack from PETA.

**Weaknesses:**

1. The most important is that the motivation of the studied problem is unclear. I doubt if there are any scenarios in reality where exists corrupted PLM trained with so much (25%) poisoned data and it needs PEFT. I suggest the authors focus on discussing the motivation in introduction.

2. I suggest the author add explanations of poisoned data to improve readability.

**Questions:**

1. The learning rate of DP is 10x of other baselines from Appendix. Besides, more epochs are used. Is there any explanation?

2. I wonder how would the poisoned rate affects the results?

---

> ### Author Response · Authors · 2023-11-21
>
> Thank you so much for taking the time to review our paper. We highly appreciate and value your feedback.
>
> To address the first weakness that you pointed out, we **updated the introduction to better delineate the motivation of our work in the revised version**. Regarding the poisoning rate, we chose to operate in the setting where a compromised pre-trained language model is released to the user because this avoids the need to release poisoned examples to the user during the model training process, which makes our attack stealthy in the sense that it is robust to existing dataset filtering methods (see “Downstream Activation” in Section 3). There are also other backdoor attacks in the NLP literature that do this [1, 2, 3, 4]. This also means that the amount of poisoned data used during the first stage (25% in our main experiments) matters only for optimizing the performance of the final model resulting from the second stage and is otherwise unimportant since it doesn’t affect stealthiness. Regarding realisticity, we consider the scenario where the attacker would like to control the behavior of a particular model for a specific application (e.g., perhaps an attacker is only interested in compromising a spam detection system). Once the attacker injects a backdoor with bilevel optimization, the PLM will be released and any user that uses the selected PEFT method and dataset will be affected. We acknowledge that it is unlikely for every user of the PLM to be affected, since they could use the PLM for any arbitrary downstream task with any training strategy, but we argue that if the goal of the attacker is to succeed a fraction of the time, then PETA will meet this goal. Additionally, **we updated our paper with experiments that test for transferability of PETA to new domains and PEFT methods during the adaptation stage (see Section 4)**, which broadens the scope of our attack.
>
> In response to the second weakness, thank you for pointing this out! We realized that some of the details related to the dataset poisoning processes were missing and **fixed this in our revised paper (see “Implementation Details” in Section 4)**.
>
> In response to the first question, DP is the only baseline that involves parameter-efficient fine-tuning (all of the other baselines in Table 12 require updating 100% of the model parameters), and for PEFT methods, we decided to use learning rates close to 2e-4 based on recommendations from relevant work on PEFT [5]. Note that we used similar learning rates for PETA as well (stated in Appendix A.4). Additionally, more epochs were required to reach convergence for DP.
>
> In response to the second question, we think that this is a good point to consider and **updated the paper with results from varying the poison rate (see “Sensitivity to Poison Rate” in Section 4)**.
>
>
> [1] https://aclanthology.org/2020.acl-main.249.pdf
>
> [2] https://aclanthology.org/2021.emnlp-main.241.pdf
>
> [3] https://arxiv.org/pdf/2110.02467.pdf
>
> [4] https://arxiv.org/pdf/2111.00197.pdf
>
> [5] https://arxiv.org/pdf/2110.04366.pdf

---

> > ### Author Response · Authors · 2023-11-22
> > **Additional feedback**
> >
> > We wanted to follow up and see if the previous responses and revisions addressed your concerns. We would be happy to answer any additional questions that you may have.
> >
> > Thanks again for helping us improve our paper!

---

> > > ### Comment · Reviewer_tYon · 2023-11-23
> > >
> > > Thanks for your response. I am happy to increase my score to 6.

---

### Official Review · Reviewer_EEAs · 2023-10-30

**Soundness:** 3 good
**Presentation:** 3 good
**Contribution:** 2 fair
**Rating:** 6
**Confidence:** 5

**Summary:**

This paper focuses on trojan/backdoor attacks in the parameter efficient fine-tuning (PEFT) setting, which is an important research topic. The authors propose PETA, a novel attack to inject the backdoor into the PLM using bilevel optimization. Extensive experiments demonstrate the effectiveness of PETA. The authors also discuss potential countermeasures.

**Strengths:**

- important topic
- well-written paper
- effective attacks and countermeasures

**Weaknesses:**

- technical novelty is limited
- unclear attack description
- more evaluations are needed

**Questions:**

- My main concern is that the proposed attack leverages existing backdoor attack methodology in the scenario of PEFT, making the technical novelty limited. Please correct me if I am wrong. In both Eq. 2 and Eq. 3, the whole model's parameters seem to be updated (including $\theta$ and $\delta$), which is the same as the backdoor attack in the fine-tuning stage. After the attack, $\delta$ will be discard and a new $\delta$ will be trained (with $\theta$ being fixed) by the victim user to perform the downstream task. I appreciate it if the authors could better clarify the advantage of the proposed attack compared to previous attacks and discuss the attack process more clearly.

- Regarding the evaluation, it seems that BadNet can also achieve both high ACC and high LFR. Would it be the case if we discard the classifier $\delta$ backdoored by BadNet and train a new $\delta$?

- During the backdoor process, the dataset is from the same distribution as the testing data, which is a relatively strong assumption. I would suggest the authors also evaluate the attack performance when the downstream dataset is from a different distribution than the attack dataset.

- I like the authors' idea regarding the defense. As shown in Fig. 3, the LFR can be largely reduced if we could select the optimal layer. However, it would be hard to select the optimal one. Previous work[a] also suggests that fine-tuning the whole model could be an effective defense. Would it be possible to make a trade-off by fine-tuning the last few layers?

[a] https://arxiv.org/abs/2212.09067

---

> ### Author Response · Authors · 2023-11-21
>
> Thank you so much for taking the time to review our paper. We highly appreciate and value your feedback.
>
> Based on your comments, we **ran additional experiments to check for domain transferability and found that PETA is able to generalize (see “PETA transfers to new domains” in Section 4)**. We also tested for transferability to new PEFT methods and **updated the paper with the results (see “PETA transfers to unseen PEFT methods” in Section 4)**.
>
> Further, we tested the defense idea that was suggested and **updated Section 6 with our findings**. Thank you so much for showing us this reference!
>
> The variation of BadNet that was mentioned in Question 2 seemed like an interesting baseline to consider, so we conducted these experiments on SST-2, Offenseval, and AG’s News, and observed clean accuracies and label flip rates (shown below) that were very similar to the results of standard BadNet, so we did not add them to the paper.
>
> | Dataset | ACC | LFR |
> | -------- | ------- | ------- |
> | SST-2 | 92.92 | 100 |
> | Offenseval | 84.75 | 100 |
> | AG’s News | 91.76 | 96.05 |
>
>
> Regarding your first question, we leverage existing triggers in the backdoor attack literature during evaluation to demonstrate that our PETA attacks can work on all types (both insertion-based and hidden triggers). However, the novelty of our work lies in the attack framework, which can conceptually be used with any trigger and is designed specifically for PEFT (the bilevel optimization would not be possible outside of this paradigm). We are one of the first to propose backdoor attacks targeting PEFT and are the first to devise an approach for poisoning the frozen PLM by exploiting the unique characteristics of PEFT. For a more in-depth discussion of our motivation and connections to related work, **please see our revised paper (see Section 1 and “Connection to Other Work” in Section 3)**. Regarding the attack process, your interpretation is correct and we have **updated the paper to include a more detailed description of the training algorithm for the bilevel optimization phase of the attack (see “Training Algorithm” in Section 3)**.

---

> > ### Comment · Reviewer_EEAs · 2023-11-22
> >
> > Thanks for the clarification, I would raise the score to 6.

---

> > > ### Author Response · Authors · 2023-11-22
> > >
> > > Thank you so much!

---

### Official Review · Reviewer_oWxS · 2023-10-31

**Soundness:** 2 fair
**Presentation:** 3 good
**Contribution:** 3 good
**Rating:** 5
**Confidence:** 4

**Summary:**

This work reveals that Parameter-efficient fine-tuning (PEFT) exhibits unique vulnerability to trojan attacks. A novel attack called PETA was presented that accounts for downstream adaptation through bilevel optimization: the upper-level objective embeds the backdoor into a PLM while the lower-level objective simulates PEFT to retain the PLM’s task-specific performance. Extensive evaluation across a variety of downstream tasks and trigger designs demonstrate PETA’s effectiveness in terms of both attack success rate and unaffected clean accuracy, even after the victim user performs PEFT over the backdoored PLM using untainted data.

**Strengths:**

1. The proposed trojan attack under Parameter-efficient fine-tuning (PEFT) setting is interesting and practical.
2. the experimental results seems promising.
3. The paper is generally well motivated and written.

**Weaknesses:**

1. How complex is the bilevel optimization?
2. Baselines are all old ones back to 2-5 years before.
3. The defense part is bit weak, just considered a simple one.
4. Some highly relevant works on backdoors are missing:
Fine-mixing: Mitigating Backdoors in Fine-tuned Language Models
Backdoor attacks on self-supervised learning
ASSET: Robust Backdoor Data Detection Across a Multiplicity of Deep Learning Paradigms
Reconstructive Neuron Pruning for Backdoor Defense
Anti-Backdoor Learning: Training Clean Models on Poisoned Data
Neural Attention Distillation: Erasing Backdoor Triggers from Deep Neural Networks

**Questions:**

see the Weaknesses.

---

> ### Author Response · Authors · 2023-11-21
>
> Thank you so much for taking the time to review our paper. We highly appreciate and value your feedback.
>
> To address the first weakness, we **updated Section 3 with a detailed description of the training algorithm (see “Training Algorithm”)**.
>
> To address the fourth weakness, we **updated Section 2 to include all of the works**. Thank you for pointing this out!
>
> In response to the second weakness, we included six baselines (excluding DP attacks, which were introduced in our work): BadNet (Gu et al., 2017), LWP (Li et al., 2021a), POR (Shen et al., 2021), AddSent (Dai et al., 2019), StyleBkd (Qi et al., 2021b), and SynBkd (Qi et al., 2021e). Since PETA can conceptually work on any type of trigger, we selected the sentence trigger from the AddSent attack, the style trigger from the StyleBkd attack, and the syntactic trigger from the SynBkd attack, and included the original attacks as baselines to make direct comparisons. To the best of our knowledge, AddSent/SynBkd/StyleBkd are the latest sentence/syntactic/style attacks that are reproducible and update 100% of the parameters. LWP and POR both release backdoored pre-trained language models and were selected because they fall in this category and are among the latest works as well. Lastly, BadNet was selected because it was the first backdoor attack (initially for computer vision) and serves as a strong point of comparison. For these reasons, we believe that the baselines represent the state of the art of backdoor attacks against LLMs.
>
> In response to the third weakness, we acknowledge that the proposed defenses aren’t full-blown defenses due to the absence of an algorithm that selects the optimal layer, but if the defender were to randomly choose, they would be able to defend against the attack reasonably well with high probability. From the figures, we can see that there are usually many choices of layers that lead to low LFRs (e.g., any layer between 7 and 10 would bring the LFR down to at most 30% for PETA-Adapters-Syntax and **unfreeze_attn** from Figure 4). We would also like to point out that **Section 6 has been updated to include results for another defense strategy** that was suggested by a different reviewer, which further demonstrates the efficacy of our initial defender.
>
> Lastly, we wanted to mention that **we updated our paper with experiments that test for transferability of PETA to new domains and PEFT methods during the adaptation stage (see Section 4)**, which broadens the scope of our attack.

---

> > ### Author Response · Authors · 2023-11-22
> > **Additional feedback**
> >
> > We wanted to follow up and see if the previous responses and revisions addressed your concerns. We would be happy to answer any additional questions that you may have.
> >
> > Thanks again for helping us improve our paper!

---

> > > ### Comment · Reviewer_oWxS · 2023-11-23
> > > **Official Comment by Reviewer oWxS**
> > >
> > > Thank you for your response. After careful consideration, I still think this paper has large rooms to improve in order to catch up with the standards of ICLR, so I keep my original score.

---

### Author Response · Authors · 2023-11-22
**Response to all reviewers**

We would like to thank all of the reviewers again for their helpful feedback. We responded to each reviewer individually, but would like to give a summary of all of the major revisions that were made in the updated version of our paper:

- Section 1 was updated to better delineate our motivation
- Section 2 was updated to include additional related work
- Section 3 was updated with detailed descriptions of the training algorithm and connection to other work
- Section 4 was updated to (1) include new results on generalization to unseen domains and PEFT methods during the downstream adaptation stage; (2) include a better description of the dataset poisoning processes; and (3) discuss experiments from varying the poison rate
- Section 6 was updated to include results for another defense strategy

---

### Meta-Review · Area_Chair_xWsh · 2023-12-10

**Metareview:**

In this paper, the authors introduced PETA, a backdoor attack that is designed specifically for the parameter-efficient fine-tuning paradigm. Through extensive experiments, PETA can outperform standard dataset poisoning attacks when applied in the PEFT setting and perform comparably to its attack counterparts that update 100% of the parameters. PETA also possesses the ability to transfer to new domains and PEFT techniques during the adaptation stage. Lastly, the authors explored a defense mechanism specifically for PETA and revealed the importance of self-attention for encoding backdoor information in higher layers.

The paper is well-written and easy to follow. The experiments are well set-up and the results are good. However, the novelty of the paper is somewhat limited. During the rebuttal, the reviewers had several questions about the technical details and potential impact of the work, and the authors addressed them well in the response.

**Justification For Why Not Higher Score:**

The novelty of the paper is somewhat limited

**Justification For Why Not Lower Score:**

N/A

---

### Decision · Program_Chairs · 2024-01-16

Reject